# Forest fire threatens global carbon sinks and population centres under rising atmospheric water demand

Hamish Clarke [1,2,3,4] ✉, Rachael H. Nolan [2,3], Victor Resco De Dios [5,6,7], Ross Bradstock[1,2,8], Anne Griebel[2,3], Shiva Khanal[3] & Matthias M. Boer [3]

Levels of fire activity and severity that are unprecedented in the instrumental record have recently been observed in forested regions around the world. Using a large sample of daily fire events and hourly climate data, here we show that fire activity in all global forest biomes responds strongly and predictably to exceedance of thresholds in atmospheric water demand, as measured by maximum daily vapour pressure deficit. The climatology of vapour pressure deficit can therefore be reliably used to predict forest fire risk under projected future climates. We find that climate change is projected to lead to widespread increases in risk, with at least 30 additional days above critical thresholds for fire activity in forest biomes on every continent by 2100 under rising emissions scenarios. Escalating forest fire risk threatens catastrophic carbon losses in the Amazon and major population health impacts from wildfire smoke in south Asia and east Africa.

Earth's forests and woodlands have been marked by a string of mega-fires in recent years[1]. Impacts on humans and ecosystems extend well beyond the footprint of these fires[2], which are increasingly occurring in areas and seasons not normally considered fire-prone[3]. With their abundance of live and dead plant biomass (fuel), all forests and woodlands are inherently flammable. The drying out of fuel in these vegetation communities represents a critical transition to a higher risk state, with connected dry areas forming a template for any wildfires that occur[4]. Major drying events can overcome natural barriers to fire spread such as relatively moist vegetation in gullies[5] and in extreme cases allow fire to spread into rainforests and other fire-sensitive forest communities[6,7].

Quantifying the relationship between forest fire activity and variation in fuel moisture content thus provides a means for prediction of landscape fire potential, particularly when accompanied by spatially explicit predictions of fuel moisture content. The moisture content of fine dead plant material on the forest floor is a key determinant of fire properties[8] and can be predicted from temperature and humidity, inputs for which a wide range of global, high-quality observational and modelling datasets are available[9,10]. Vapour pressure deficit (VPD), which is calculated from air temperature and humidity, is a direct measure of the atmospheric demand for water and has been shown to be a reliable predictor of dead fuel moisture content in a range of forest and woodland biomes[11]. VPD is also a key driver of plant mortality, causing declines in the moisture content of live fuels and an increase in the proportion of highly flammable dead fuels[12]. VPD-based fuel moisture thresholds have been shown to be indicative of critical increases in the cumulative area burnt in southeast Australia[13] and Mediterranean Europe[14]. VPD itself has been found to be strongly associated with fire activity in boreal[15], temperate[4,16–21], Mediterranean[22] and tropical forests[23].

[1]Centre for Environmental Risk Management of Bushfire, Centre for Sustainable Ecosystem Solutions, University of Wollongong, Wollongong, Australia. [2]NSW Bushfire Risk Management Research Hub, Wollongong, Australia. [3]Hawkesbury Institute for the Environment, Western Sydney University, Richmond, Australia. [4]School of Ecosystem and Forest Sciences, University of Melbourne, Parkville, Australia. [5]Department of Crop and Forest Sciences, Universitat de Lleida, Lérida, Spain. [6]JRU CTFC-AGROTECNIO-Cerca Center, Lérida, Spain. [7]School of Life Science and Engineering, Southwest University of Science and Technology, Mianyang, China. [8]Applied Bushfire Science Program, NSW Department of Planning, Industry and Environment, Parramatta, Australia. ✉e-mail: hamish.clarke@unimelb.edu.au

Here, we identify VPD thresholds for the switching of global forest ecosystems from a prevailing humid and non-flammable state to a dry, flammable (i.e. ignitable) state. Our use of daily remotely sensed burned area and hourly climate reanalysis data is a key advance on previous studies, which typically focus on aggregate measures such as total area burnt over a season. Given this focus, models of the probability of successful ignition as a function of climate (i.e. daily maximum VPD) should provide a better identification of the critical fuel moisture threshold than models associated with total area burnt or the incidence of large fires. The latter may be confounded by additional factors such as fire suppression or the natural variation in area burned across biomes and regions. We develop generalised linear models of the probability of fire occurrence and use these models along with

skill-selected global climate models to assess the impacts of climate change on the frequency of exceedance of fire activity thresholds (see Methods). We focus on the implications of changes in forest fire activity in two critical areas: carbon losses[24] and human health impacts from wildfire smoke[25,26].

## Results and discussion

We found that fire activity in all global forest biomes responds strongly and predictably to VPD, with a clear difference in the distribution of VPD values on fire days compared to non-fire days (Fig. 1; Supplementary Note 1). Our models performed well in most forest biomes, with a median true positive rate of 0.73 ($n = 70$), meaning the probability of correctly predicting fire on a fire day was 73% (Supplementary

**a Canada**
Boreal Forests/Taiga

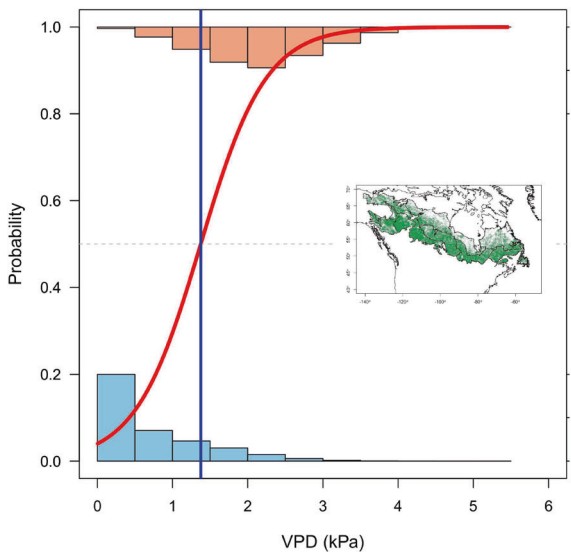

**b Australia**
Temperate Broadleaf & Mixed Forests

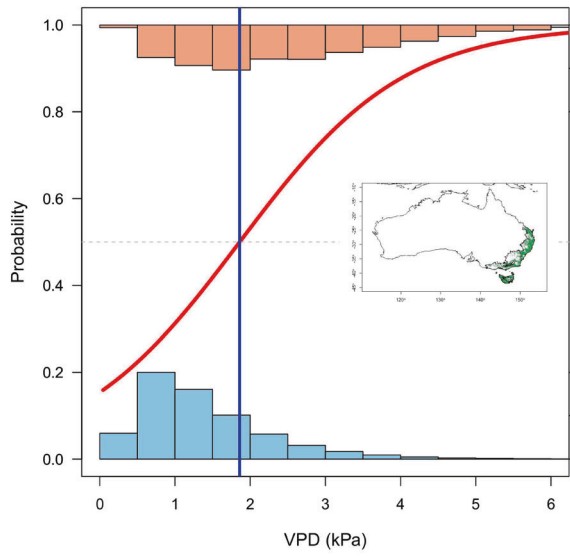

**c South America (North)**
Tropical & Subtropical Moist Broadleaf Forests

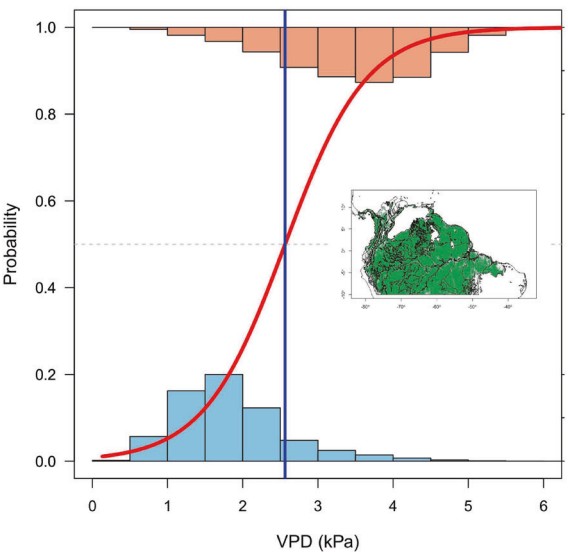

**d Europe**
Mediterranean Forests, Woodlands & Scrub

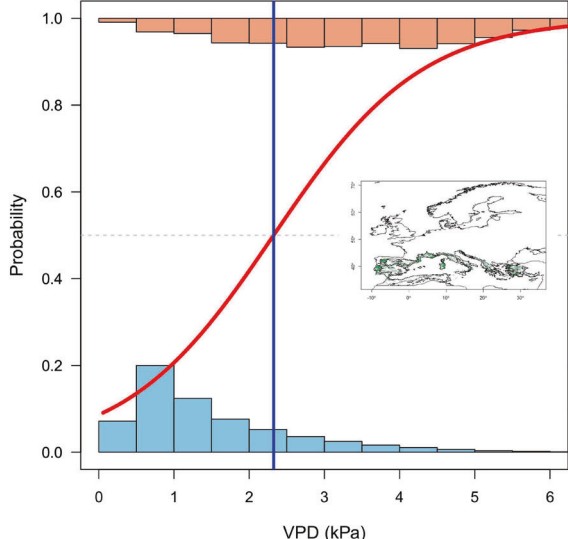

**Fig. 1 | The response of forest fire activity to VPD in four continental forest biomes. a** Boreal forests and taiga in Canada ($n = 1,657,115$). **b** Temperate broadleaf and mixed forests in Australia ($n = 580,200$). **c** Tropical and subtropical moist broadleaf forests in northern South America ($n = 1,697,491$). **d** Mediterranean forests, woodlands and scrub in Europe ($n = 147,605$). The lines show generalised

linear models of the probability of fire as a function of daily VPD (red) and the threshold at which the probability of fire is 50% (blue). Histograms show the distribution of VPD on days with fire (top) and without fire (bottom). See Supplementary Information for model performance details.

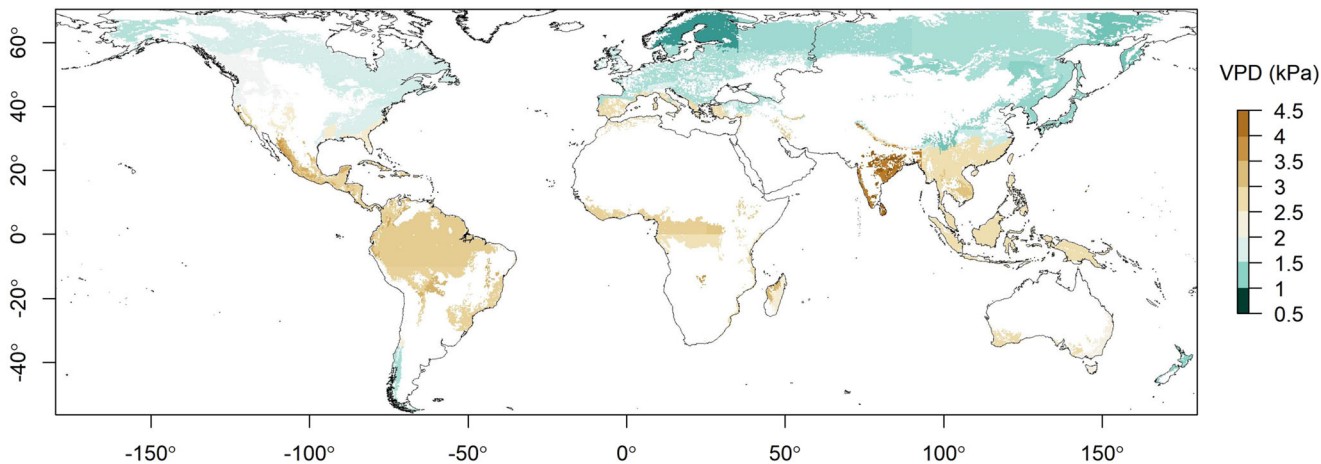

**Fig. 2 | VPD thresholds (kPa) for fire activity in global forest biomes.** Threshold values indicate the daily VPD above which the probability of fire exceeds 50%, as derived from generalised linear modelling of historical climate and fire records. The white areas indicate non-forest land.

Fig. 1; Supplementary Table 1). There were marked differences in the VPD threshold above which the daily probability of fire exceeds 50% in the different forest biomes (Fig. 2, Supplementary Fig. 2, Supplementary Table 2). These correspond broadly to latitudinal gradients, with higher thresholds—corresponding to warmer and drier conditions and hence greater evaporative demand—occurring closer to the equator and lower values occurring at higher latitudes. VPD thresholds were highest in subtropical and tropical biomes (median 2.7 kPa), followed by mediterranean biomes (median 2.3 kPa). Temperate and boreal biomes had much lower thresholds above which fires were probable (median 1.3 kPa). The mean annual frequency of daily VPD threshold exceedances (or the number of potential fire days) varied widely across forest biomes (Fig. 3a, Supplementary Tables 1 and 2).

Unlike the VPD threshold value, there was little clustering of the frequency of days exceeding the VPD threshold among forest types, nor was there a strong latitudinal gradient. Further, there was generally much greater variability within individual forest biomes for this metric than for the VPD threshold value (Supplementary Fig. 3). Between 2003 and 2020, the mean annual frequency of VPD threshold exceedances was greatest in forests of east Asia, southwest Australia, western Europe and the eastern United States. Regions where VPD thresholds occurred fewer than 30 days of the year on average were found in temperate, boreal, subtropical and tropical forest biomes. In contrast, no mediterranean forest biome exceeded its VPD threshold fewer than 66 days per year, highlighting the fire proneness of mediterranean-type forests under current climate conditions.

Unmitigated climate change is projected to lead to widespread increases in the frequency of days exceeding VPD thresholds associated with elevated probability of fire. Under a high emissions scenario (RCP8.5), by 2026–2045 all models projected at least 45 additional days per year above the VPD threshold in parts of tropical South America, with two out of three models also projecting increases of this magnitude in North America, east Africa and large parts of Europe (Supplementary Fig. 4). By 2081–2100 the magnitude of change is projected to be far greater, even in the model with the smallest increases (Fig. 3). Then VPD thresholds will be exceeded by at least 45 additional days per year in forest biomes on every continent, including increases of at least 150 days per year in tropical South America, regardless of model.

Under a lower and increasingly more plausible emissions scenario[27] the magnitude of change is smaller but still features widespread increases in the annual frequency of days of elevated probability of fire (Supplementary Fig. 5 and 6). Increases are widespread across time horizons, models and emissions scenarios, with the largest projected increases in the frequency of such days in tropical

forests, followed by northern hemisphere temperate forests and boreal forests. Although of lower magnitude, the projected increases in days exceeding fire activity thresholds in mediterranean forests occur against a backdrop of an already high annual frequency of such days. The increases are greatest and most widespread in ACCESS1-0 and GFDL-CM3 and generally more moderate in CNRM-CM5. The latter tends to project the least warming of the three models, with GFDL-CM3 projecting the most[28]. ACCESS1-0 is generally the driest of the three models, while both ACCESS1-0 and GFDL-CM3 have a higher climate sensitivity parameter than CNRM-CM5[29]. Increases in days over the VPD threshold are projected to occur in regions with globally significant forest carbon storage, including the Amazon in tropical South America and the Congo in Central Africa (Fig. 4). Substantial increases in the number of days over the VPD threshold—and hence days of elevated probability of fire and smoke emissions—are projected to occur by 2081–2100 near major population centres in south Asia and east Africa by all three models (Fig. 5). Two of three models also suggest considerable population exposure to smoke from increased forest fire activity in parts of central America, west Africa and east Asia.

We found that for many forested regions, and for the majority of global burned area in forests, the probability of fire occurrence can be accurately predicted on the basis of exceedance of thresholds in daily maximum VPD. We also found that the value of these thresholds varied predictably across major forest types, being highest in tropical and subtropical forests and lowest in temperate and boreal forests. Improving our understanding of the drivers of fuel moisture and its links to forest fire activity are critical to the development and use of predictive models[30,31]. Our findings provide new evidence at a high temporal resolution (i.e. daily) of the link between fuel moisture and forest fire activity[32,33] and the potential for fuel moisture-mediated changes—nearly always increases—in risk due to climate change[34–36]. A recent study identified VPD thresholds associated with fire activity in North and South America between 2017–2020 using hourly data, with similar findings[37]. We did not explore seasonality, interannual variability or temporal trends of atmospheric water demand in our study, but there is already evidence of increasing dryness in the Mediterranean, western US and tropical South America[12,38,39], along with increases in global forest carbon emissions[40], attributed in part to changes in fuel moisture[41]. Our use of VPD is pragmatic and we note that model performance in specific regions may be improved with the use of alternative predictors[42] (e.g. evaporation, soil moisture and wind speed) or by the aggregation of predictors[43]. Equally, other sources of meteorological and fire incidence data may provide a greater estimate of the uncertainty around these results[44], although the improved

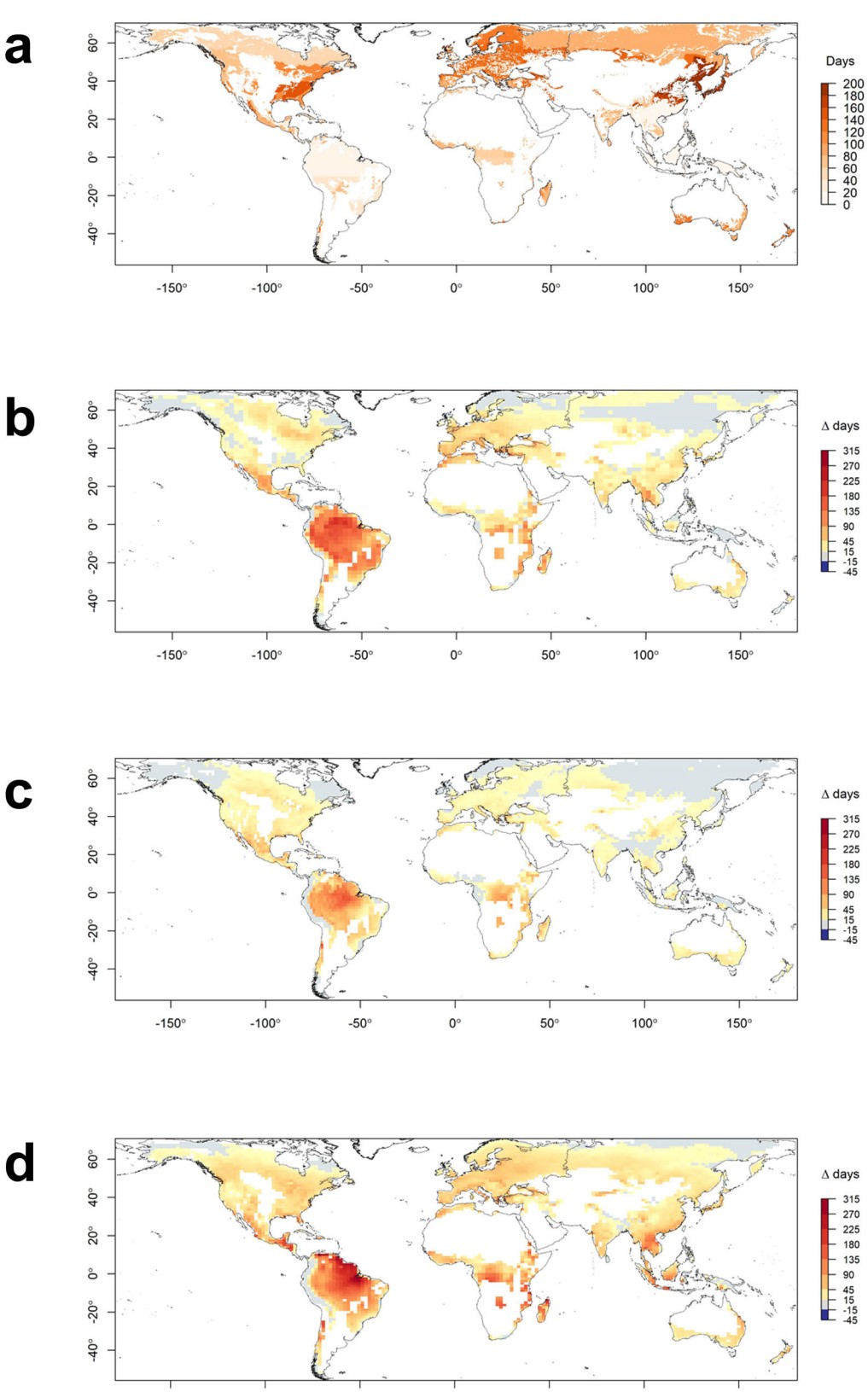

**Fig. 3 | The mean annual frequency of daily VPD threshold exceedances (days) for global forest biomes.** Current frequency based on ERA5 data (2003-2020) (**a**) and the projected change in the number of days over VPD threshold by 2081–2100 under RCP8.5 for the GFDL-CM3 (**b**), CNRM-CM5 (**c**) and ACCESS1.0 (**d**) models. The white areas indicate non-forest land.

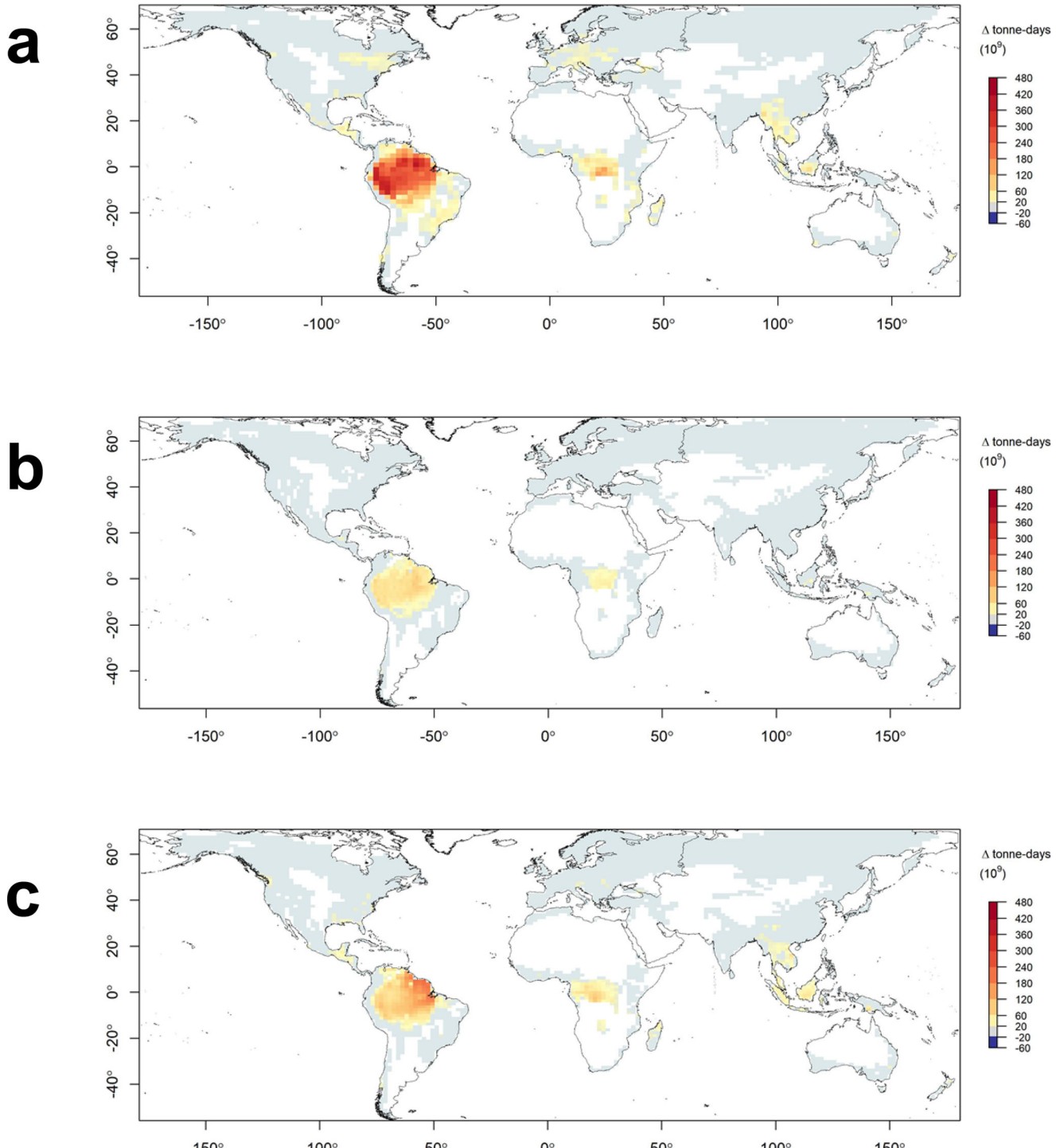

**Fig. 4 | Projected change in global forest aboveground biomass exposure to changes in the mean annual frequency of days exceeding the VPD threshold by 2081–2100 under RCP8.5.** Results are shown for GFDL-CM3 (**a**), CNRM-CM5 (**b**) and ACCESS1.0 (**c**) models. Units are tonne-days of exposure. The white areas indicate non-forest land.

representation of humidity is a noted feature of the ERA5 reanalysis[45]. The generally strong performance of our models is notable given they do not explicitly address other key biophysical constraints on fire—which may not always act synergistically with the changes in fuel moisture projected here—such as human activity[46], fire weather[47] and long-term drying[48]. High population density, high fire detection rates and high suppression capacity are all known to lower the effective ignition rate and could weaken the link between VPD and fire activity in some regions[49,50].

Increasing forest fire risk has widespread implications for humans, ecosystems and the global carbon cycle. Our analysis highlights the carbon-rich forests of tropical South America as being exposed to substantial increases in forest fire activity under climate change. At a local level, these results reaffirm the need to understand the complex and dynamic drivers and effects of fire—and fire management—in these regions[51,52]. At a global scale, our findings point to the Amazon rainforest as a "tipping element" i.e. a site for which the crossing of some critical threshold could have major consequences for the state or

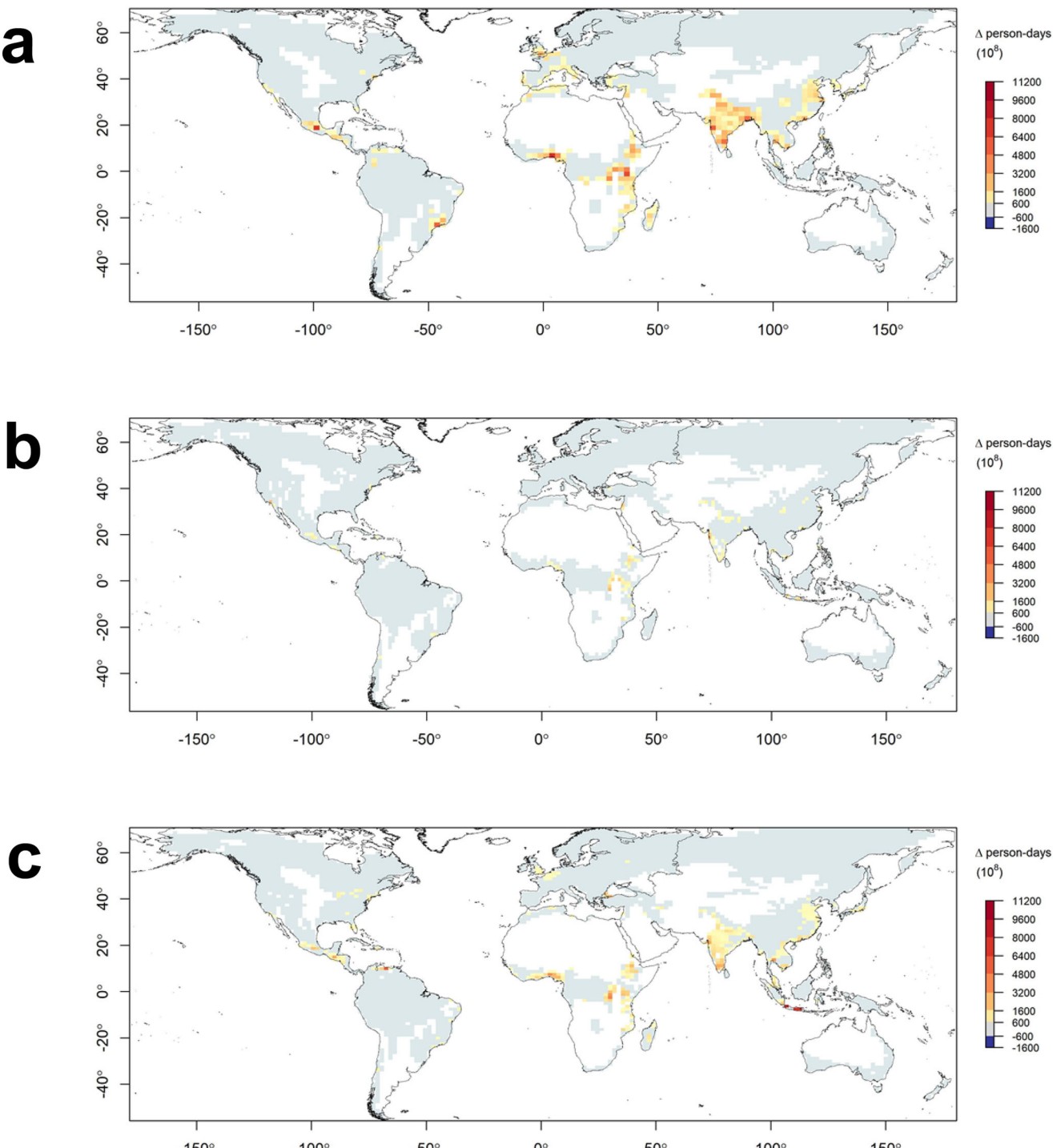

**Fig. 5 | Change in population exposure to changes in the mean annual frequency of days exceeding the VPD threshold by 2081–2100 under RCP8.5.** Results are shown for GFDL-CM3 (**a**), CNRM-CM5 (**b**) and ACCESS1.0 (**c**) models. Units are person-days of exposure. The white areas indicate non-forest land.

development of the earth's climate system[53]. There is already evidence that recent increases in fire may have tipped the Amazon from a net carbon sink to a net carbon source[54]. Increasing wildfire at the scale described here could interact with other sources of dieback such as drought and deforestation to further undermine the role that the Amazon plays within the carbon cycle and regional climate, as a contributor to human welfare and as a unique feature of the biosphere. Likewise boreal forests, another biome for which we project increases in fire activity, have also been identified as tipping elements[53]. Our findings highlight the risks posed by conditions of increasing

atmospheric moisture demand to forest-based efforts to enhance terrestrial carbon storage such as reforestation, offsetting and improved forest management[55].

We also show that increases in forest fire activity are projected to occur near major population centres in east Africa and south Asia, and possibly central America, east Asia and west Africa. These populations may be exposed to increased wildfire smoke, which can have substantial impacts on human health. There have already been significant wildfire smoke events in Russia in 2010[56] and in equatorial Asia in 2015[57], while the Australian mega-fires of 2019-20 were estimated to

have led to 429 excess deaths and a much larger number of hospitalisations due to wildfire smoke[58]. Overall the health costs of the 2019–2020 Australian fires were close to US$1.5 billion[58], a number comparable to one estimate of the annual acute health impacts from wildfire smoke in Canada[59]. We have modelled forest fire only, and any increases in fire risk that extend to peatlands will lead to even greater health impacts[57]. While significant, smoke health costs represent just a fraction of the broader economic impacts, reflecting the wide range of direct and indirect effects of fire including on property, infrastructure, agriculture and tourism. Our study provides tangible evidence of the local, regional and global impacts of forest fire under future climates that may be avoided by successfully mitigating anthropogenic climate change.

## Methods

### Study area
Our study area consisted of all global forest biomes. We first selected forest-dominant biomes from a global classification of terrestrial ecosystems[60]. The resultant biomes formed three major groups: subtropical and tropical (Tropical and Subtropical Moist Broadleaf Forests, Tropical and Subtropical Dry Broadleaf Forests, Tropical and Subtropical Coniferous Forests); mediterranean (Mediterranean Forests, Woodlands, and Scrub) and temperate and boreal (Broadleaf and Mixed Forests, Temperate Coniferous Forests, Boreal Forests/Taiga). These biomes were then masked using a 1 km resolution global forest cover product in order to further resolve forests[61]. Selected study area properties are shown in Supplementary Table 3.

### Fire data
Fire activity was represented using the Moderate Resolution Imaging Spectroradiometer (MODIS) MCD64A1 burned area product (Collection 6)[62]. We analysed fires occurring from January 1, 2003 to February 29, 2020, coinciding with the end of the austral summer associated with the extraordinary fires of 2019-20. We only used data with the highest quality assessment (QA) ratings. These data are at approximately 500 m resolution with daily timestep. In order to explore variation within biomes we used 21 pre-defined sub-continental windows[63] (Supplementary Table 4). Windows 22, 23 and 24, corresponding to the Azores, Cape Verde Island and Hawaii, were omitted. Although they are of great interest, prescribed and cultural burns are not likely to have accounted for a significant proportion of the fire activity data as they are generally of far lower size and intensity than wildfires and are frequently undetected by MODIS[64].

### Climate data
We computed daily vapour pressure deficit (VPD) using daily maximum air temperature and dew point temperature at the time of daily maximum air temperature, based on data from the ERA5 reanalysis[45], for the same period as the burned area data. The ERA5 data has a horizontal resolution of 0.25° and hourly temporal resolution. For climate change analyses we selected three global climate models from the CMIP5 dataset[65] on the basis of skill, independence and the ability to span the range of future changes in climate: ACESS1.0, CNRM-CM5 and GFDL-CM3 (Supplementary Table 5). These models were among the best performing compared to other CMIP5 models in a comprehensive evaluation for the purposes of downscaling over multiple regions, which included annual cycles of rainfall and temperature, general circulation patterns, teleconnections and the south east Asian monsoon[66]. Of the highly performing models evaluated, these three models generally spanned all or most of the range of projected future seasonal and regional changes in climate (Supplementary Fig. 7). We avoided models from the same model family to avoid duplication of models with similar biases. We used the RCP4.5 and RCP8.5 greenhouse gas emission concentration pathways, which represent

'stabilisation without overshoot' and 'rising' pathways respectively[66]. Daily maximum VPD was computed using daily maximum air temperature and relative humidity at the time of maximum air temperature from 3-hourly GCM data for the time periods 2026-2045 (mid-century) and 2081-2100 (late century). Daily ERA5 data (1981-2000) was used to bias correct GCM VPD following a quantile mapping approach[67]. Climate change values were calculated using the delta method i.e. by subtracting modelled present values (1981-2000) from modelled future values. The native resolution of the climate models was retained for the analysis, meaning that results only apply to forest within a given climate model grid cell.

### Analysis
To examine the influence of daily maximum VPD on the probability of wildfire we used a generalised linear model with binomial error distribution and logit link function. For each combination of forest biome and sub-continental window ($n = 70$) we estimated the probability of fire incidence (i.e. a grid cell being recorded as burnt) as a function of daily VPD. Presence data were the VPD values on the same day and closest grid cell to each MODIS burnt area grid cell. Due to a mismatch between the spatial resolution of fire and climate data, the same VPD value may be assigned to multiple burned area grid cells within a single climate grid cell. Quasi-absence data was generated by randomly sampling unburned grid cells within the study area at random dates throughout the year[68]. An equal number of presence and absence data was used each year and overall. A supplemental analysis confirms that presence and absence data points are drawn from the same climate zone (Supplementary Fig. 8). Grid cells that had burned in the last five years were excluded from the analysis. We set the critical forest fire activity threshold as the daily VPD value above which the probability of fire is 50% ($VPD_{P=50}$). Uncertainty in $VPD_{P=50}$ was initially represented using confidence intervals ($\pm 2 \times$ standard error). However, as confidence intervals were narrower than $\pm 0.01$ in 68 of 70 cases these figures were not reported. The area under the curve (AUC) of the receiver operating characteristic (ROC) plot was used to measure each model's prediction accuracy[69]. A discussion of model performance including accuracy and percentage deviance explained can be found in the Supplementary Information. For each combination of forest biome and sub-continental window, climate model, emissions scenario and epoch we calculated the annual frequency of days exceeding $VPD_{P=50}$. We used ERA5 data to estimate the current frequency of such days and the CMIP5 data to calculate their future frequency. Note that the strength of the relationship between VPD and fire activity in any given region does not imply a particular magnitude of burnt area for a given number of exceedances of daily VPD threshold values. A supplemental analysis examined the relationship between area averaged monthly days over $VPD_{P=50}$ and burnt area, with broadly similar findings to the main analysis (Supplementary Fig. 9). A supplemental analysis examined the relative, rather than absolute, change in the number of days over $VPD_{P=50}$, with broadly similar findings (Supplementary Figs. 10–13). All data analysis was carried out in R[70]. To estimate the potential impact of smoke exposure to human populations in the vicinity of areas that exceed the VPD threshold, we used gridded spatial demographic projections at 1 km resolution[71]. Because results are reported at the coarser resolution of GCM grid cells, they allow for long range smoke transport, which has been observed around the world at a scale of hundreds of kilometres or more[57,72]. Population projections for 2090 were based on a "middle of the road" scenario in terms of expected population growth, urbanisation, and spatial patterns of development[71]. We multiplied the population density by the change in days per year above $VPD_{P=50}$ to produce a gridded raster of the annual number of person-days of exposure to critical fire activity conditions[73]. A similar approach was taken to estimate potential

forest carbon exposure to fire. We used the tiled 100 m spatial resolution ESACCI aboveground biomass (AGB) rasters for year 2010 on a global coverage[74]. To derive an AGB raster of comparable resolution to other input datasets, the grid cells in AGB rasters were resampled with the median rule using ERA5 data as a template. While resampling, missing data areas (AGB = 0 values) were omitted. We multiplied the aboveground biomass by the change in days per year above $VPD_{P=50}$ to produce a gridded raster of the annual number of tonne-days of exposure to critical fire activity conditions. Although the exposure units (tonne-days, person days) are somewhat artificial, they transparently reflect the joint occurrence of increased frequency of high fire risk days and high density of forest carbon and human population.

## Data availability

MODIS data is available from the Land Processes Distributed Active Archive Center (LPDAAC) at the U.S. Geological Survey (USGS) Earth Resources Observation and Science Center (EROS) (http://lpdaac.usgs.gov) and the University of Maryland. ERA5 data is available from the Copernicus Climate Change Service (C3S) Climate Data Store https://cds.climate.copernicus.eu. Aboveground biomass is available from the Centre for Environmental Data Analysis https://catalogue.ceda.ac.uk. Population data is available from the National Center for Atmospheric Research (NCAR) and University Corporation for Atmospheric Research (UCAR) Climate and Global Dynamics https://www.cgd.ucar.edu/iam/modeling/spatial-population-scenarios.html. Biome data is available from the World Wildlife Fund https://www.worldwildlife.org/publications/terrestrial-ecoregions-of-the-world. The forest mask is available from Geo-Wiki https://application.geo-wiki.org/branches/biomass/. CMIP data are available from https://esgf-node.llnl.gov/search/cmip5/.

## Code availability

Code to fit the generalised linear models is available on request.

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

## Acknowledgements

The authors acknowledge the New South Wales Government's Department of Planning, Industry & Environment for providing funds to support this research via the NSW Bushfire Risk Management Research Hub. We acknowledge the World Climate Research Programme's Working Group on Coupled Modelling, which is responsible for CMIP, and we thank the climate modelling groups for producing and making available their model output. For CMIP the U.S. Department of Energy's Program for Climate Model Diagnosis and Intercomparison provides coordinating support and led development of software infrastructure in partnership with the Global Organization for Earth System Science Portals. Some of the analysis was carried out on the National Computational Infrastructure (NCI) which is supported by the Australian Commonwealth Government.

## Author contributions

M.M.B. and H.C. conceived the research. H.C. conducted the analysis and M.B., S.K., A.G. and R.H.N. contributed to it. All authors (H.C., R.H.N., V.R.D.D., R.B., A.G., S.K., M.M.B) contributed to the interpretation of the results. H.C. wrote the manuscript and all authors (H.C., R.H.N., V.R.D.D., R.B., A.G., S.K., M.M.B) reviewed and edited the manuscript.

## Competing interests

The authors declare no competing interests.
