## [Peer Review File · Nature Communications]

Forest fire threatens global carbon sinks and population centres under rising atmospheric water demandREVIEWER COMMENTS

Reviewer #1 (Remarks to the Author):

General Comments: I generally agree with the notion of increased fire potential in fuel-rich forested systems with increased daily and-seasonal atmospheric aridity. This study adds to the literature on that topic, and does so at a global scale which is commendable. I reviewed this paper for another journal a couple months ago, and the version here looks fairly similar. I'll reiterate that a novel piece of the current work is the effort to identify thresholds of daily VPD that favor fire activity in forests. Although numerous other factors (e.g. longer term fuel dryness, fire-weather, ignitions) clearly influence fire potential, the identification of robust thresholds would allow for the sort of quantification of future risk. Below I highlight some of the technical concerns and potential ways that they might be addressed.

Specific comments/considerations

1. One of the key strengths of this work is the use of a huge sample of fire events on daily timescales globally. I would suggest really highlighting this given that most global climate-fire studies look at monthly or seasonal relationships. However, the sampling approach for quasi-absence data seems off. I would suggest sampling from within the local fire season to avoid conflating weather (i.e., VPD) that is occurring during a different time of the year.
2. The authors should make a better effort to state why they use VPD rather than other variables. For example, Brey et al., (2021) contends that VPD shows a much larger increase in fire potential vs other variables (e.g., fuel moisture, fire danger). It might help to argue why VPD is a better proxy than other factors both in a contemporary perspective – and under future climate scenarios – since VPD is looked at here exclusively. That said, there is precedent for using VPD or temperature in forward looking studies. For example, Gutierrez et al., (2021) found that daily maximum temperature (and VPD) nonlinearly increased fire risk and used such projections to estimate future fire activity.
3. The methods of the paper have many potential holes that need to be addressed including
 - 3.1. Fire-VPD analyses:
 - 3.1.1. The MODIS burned area data provides a daily stamp of when the 500-m pixel burned. In the analysis, how do you deal with data independence? Namely, during large fire events, you may have an entire ERA-5 cell with fire on a given day. Would you count ~3000 cells with identical VPD values?
 - 3.1.2. Is there any way to demonstrate that you are sampling from the same climate-niche as your fire pixels? I think you are fine, unless for some reason fire pixels happened in a significant different climatological subregion. You could in a supplemental analysis show that the locations or climate of locations for presence and quasi-absence are effectively similar to counter this.
 - 3.2. Treatment of GCM data:
 - 3.2.1. While I am glad to see the authors used more than 1 GCM, I don't consider 3 GCMs to be a particularly robust result. While there is a statement about "skill selected global climate models", I don't think the reference looked at VPD. Why not use more here? Typically for climate change assessment you want to use at least 10 GCM ensembles.
 - 3.2.2. Each GCM will have biases though, so you'd ideally want to perform bias correction to get comparable VPD. The delta bias-correction approach for treating GCM data would be fine if the daily distributions of VPD from GCMs credibly represented those from ERA-5. It is unlikely that they are though.

While delta bias-correction is OK for many climate change assessments, I am concerned here given that use of thresholds from ERA-5 data were used. One would ideally want to use a more sophisticated BC approach here to account for potential large differences in the distribution.

3.3. Exposure data

3.3.1. Smoke impacts can spread well downwind of fires. In the absence of using a smoke dispersion model, it would be good to have strong justification for a distance from potential fires. There is reference to "GCM allowing smoke transport for tens of kilometers", which doesn't seem very logical as GCM resolutions are often 100-200km. Again, it might be useful to point to literature on wildfire smoke impacts here to justify choices for population exposure.

3.3.2. Did you use projected population density data or leave things at the 1990-2000 levels?

4. Figures

4.1. Fig 1: Perhaps include an inset map in each plot to show the biome of interest

4.2. Fig 3-5: The colormap here is not ideal as light blue covers increases of up to 30-days. Consider using a colormap with white / grey centered around a change of 0.

1. Guiterrez, A. et al. Wildfire response to changing daily temperature extremes in California's Sierra Nevada. *Sci. Adv.* 7, eabe6417 (2022).
2. Brey, S. J., Barnes, E. A., Pierce, J. R., Swann, A. L. S. & Fischer, E. V. Past Variance and Future Projections of the Environmental Conditions Driving Western U.S. Summertime Wildfire Burn Area. *Earth's Futur.* 9, e2020EF001645 (2021).

Reviewer #2 (Remarks to the Author):

I reviewed the paper titled "Forest fire threatens global carbon sinks and population centres under rising atmospheric water demand" by Hamish Clarke and colleagues. The authors use logistic regression models to relate the probability that a pixel burned to daily vapor pressure deficit (VPD) for 70 different regions across the globe. These models were then used to identify a VPD threshold at which the probability of fire >50%; the number of days exceeding this threshold was then calculated for each of the 70 regions. Next, the authors used climate change models to project the change in the number of days exceeding this threshold in the future. Finally, projected changes in the number of days exceeding the VPD threshold were intersected with maps of carbon and human population density to infer impacts to carbon and humans.

This is an interesting paper, but I do have concerns about whether or not the number of days exceeding the VPD threshold is actually related to area burned among the 70 regions analyzed. This is not presented, and when I qualitatively look at the results and compare them to my mental image of fire prone areas, I am not convinced there is a relationship. A little less qualitative: in looking at the animation here (https://earthobservatory.nasa.gov/global-maps/MOD14A1_M_FIRE), Japan and the Korean peninsula have very little fire, yet according to figure 3a, it has by far the highest number of days exceeding the VPD threshold. The same can be said for parts of Europe such as Scandinavia. A more complete feedback can be found below.

Major concerns:

1. It is not clear how to interpret the results pertaining to the number of days exceeding the VPD threshold. On the surface, I'd think that areas of the globe with high values (in fig. 3a) would be exceptionally fire prone and exhibit high amounts of area burned. However, I don't consider Japan and the Korean peninsula particularly fire prone. Same goes for northern Europe. In the southeastern USA, the frequency of VPD exceedance is high, but most fires are prescribed fires.
2. Related and very important: if the frequency of days exceeding the VP threshold and area burned are not at least moderately correlated, and projections of effects to carbon or people under a future climate are potentially suspect. I guess I'd like to see some sort of analyses that relates the frequency of VPD exceedance to area burned (by sub-continental window, for example). If this relationship is moderately strong, then there is reason to make the projections under climate change.
3. I'm guessing that a lot of the fire seen in some parts of the planet are cultural, agricultural, or prescribed fires. If this is the case, can projections like this even be made? Related, for those fires that are cultural/agricultural/prescribed, they probably serve to stabilize carbon, meaning these fires are generally intended to preserve large trees and not kill them. So many fires in these areas are not necessarily a threat to carbon, now and into the future.

Moderate concerns:

1. Scientists are increasingly being criticized for exaggerating the effects of climate change by, for example, using the most extreme climate change scenarios in their analyses. It is my understanding that RCP 8.5 is unlikely, so it is perhaps more appropriate to use a more relevant emissions scenario for the main findings. I know it is less splashy, but I think it is important to not overexaggerate climate change effects in the abstract and the main figures in the paper. Sure, keep RCP 8.5, but that might be better in the supplemental as opposed to the main findings.
2. The absolute change in the number of days exceeding the VPD threshold under climate change is interesting, but I am wondering about a relative metric of projected change. For example, I can envision some parts of the boreal forest increasing from 10 to 30 days, which is 20 day increase but a three-fold increase. Will the implications of this differ when compared to a projected increase from 100 to 120 days in other parts of the globe?

Response to reviewer comments – Forest fire threatens global carbon sinks and population centres under rising atmospheric water demand

Note that line numbers quoted in the response refer to the manuscript with tracked changes showing all markup.

Reviewer 1 Comment 1 (R1.1)

One of the key strengths of this work is the use of a huge sample of fire events on daily timescales globally. I would suggest really highlighting this given that most global climate fire studies look at monthly or seasonal relationships.

Response

Revised. L25-28, L62-65, L167-170. Thank you for raising this point. We agree that this aspect was not emphasised in the original aspect and have added text to the abstract (L25-28), introduction (L62-65) and discussion (L167-170) in order to further emphasise this aspect of the study.

L25-28: “Using a large sample of daily fire events and hourly climate data, here we show that fire activity in all global forest biomes responds strongly and predictably to exceedance of thresholds in atmospheric water demand, as measured by maximum daily vapour pressure deficit.”

L62-65: “Our use of daily remotely sensed burned area and hourly climate reanalysis data is a key advance on previous studies, which typically focusing on aggregate measures such as total area burnt over a season.”

L167-170: “Our findings provide new evidence at a high temporal resolution (i.e. daily) of the link between fuel moisture and forest fire activity^{30,31} and the potential for fuel moisture-mediated changes – nearly always increases – in risk due to climate change³²⁻³⁴.”

R1.2

However, the sampling approach for quasi-absence data seems off. I would suggest sampling from within the local fire season to avoid conflating weather (i.e., VPD) that is occurring during a different time of the year.

Response

Not revised. We understand this comment as many studies have restricted their analysis to ‘the fire season’. Notwithstanding the lack of consensus on precisely how to define the fire season, our intention is to model the relationship between atmospheric dryness and fire activity and to do this objectively we do not wish to bias our sample by excluding data unnecessarily and in particular by restricting our analysis to the relatively short satellite record over which fires have been observed.

Additionally, we would like to stress the important point that many studies indicate that the fire season has already lengthened in many areas globally under anthropogenic climate warming (Jolly et al. 2015) and the trend is expected to continue. In other words, the current fire season is a dubious proxy for the future fire season. Our approach does not assume that fire will remain within historical windows and has the further advantage of setting a baseline against which to detect future changes in the timing and duration of fire activity.

References

Jolly WM, Cochrane MA, Freeborn PH, Holden ZA, Brown TJ, Williamson GJ, Bowman DMJS. (2015) Climate-induced variations in global wildfire danger from 1979 to 2013. *Nature Communications* 6, 7537. <https://doi.org/10.1038/ncomms8537>

R1.3

The authors should make a better effort to state why they use VPD rather than other variables. For example, Brey et al., (2021) contends that VPD shows a much larger increase in fire potential vs other variables (e.g., fuel moisture, fire danger). It might help to argue why VPD is a better proxy than other factors both in a contemporary perspective – and under future climate scenarios – since VPD is looked at here exclusively. That said, there is precedent for using VPD or temperature in forward looking studies. For example, Gutierrez et al., (2021) found that daily maximum temperature (and VPD) nonlinearly increased fire risk and used such projections to estimate future fire activity.

Response

Revised. L169, L170. We have strengthened the case for VPD by adding references in the discussion to Gutierrez et al. (2021) and other recently published evidence (Cawson et al. 2022) linking VPD to ignition likelihood in wet forests. However, we note that the original manuscript contains extensive justification for the use of VPD (L50-58; 14 citations) and acknowledgement of alternatives (L174-177; 2 citations).

References

Gutierrez AA, Hantson S, Langenbrunner B, Chen B, Jin Y, Goulden ML, Randerson JT. (2021) Wildfire response to changing daily temperature extremes in California's Sierra Nevada. *Sci Adv*, 7(47), eabe6417. doi:10.1126/sciadv.abe6417.

Cawson, J.G., Pickering, B.J., Filkov, A.I., Burton, J.E., Kilinc, M., Penman, T.D. (2022). Predicting ignitability from firebrands in mature wet eucalypt forests, *Forest Ecology and Management*, 519, <https://doi.org/10.1016/j.foreco.2022.120315>.

R1.4

The MODIS burned area data provides a daily stamp of when the 500-m pixel burned. In the analysis, how do you deal with data independence? Namely, during large fire events, you may have an entire ERA-5 cell with fire on a given day. Would you count ~3000 cells with identical VPD values?

Response

Revised. L498-500. Thanks for raising this point. We have clarified in the methods that this is indeed the case.

“Due to a mismatch between the spatial resolution of fire and climate data, the same VPD value may be assigned to multiple burned area grid cells within a single climate grid cell.”

Ultimately one of the constraints on our analysis is the spatial and temporal resolution of the climate and fire data. While this results in unsampled variation for the averaging region and time period (as in your example) our results show that there are still strong regional patterns in the relationship between VPD and fire activity.

R1.5

Is there any way to demonstrate that you are sampling from the same climate niche as your fire pixels? I think you are fine, unless for some reason fire pixels happened in a significant different climatological subregion. You could in a supplemental analysis show that the locations or climate of locations for presence and quasi-absence are effectively similar to counter this.

Response

Revised. L502-504, Supplementary Fig. 8. We have added a summary of the correlation between the climate of presence and absence data using independent very high resolution climate data (Fick & Hijmans 2017). The high correlations provide confidence that these data have been sampled from the same climate niche. Please refer to Supplementary Fig. 8, reproduced below. Text has been added to the methods as follows:

“A supplemental analysis confirms that presence and absence data points are drawn from the same climate zone (Supplementary Fig. 8).”

References

Fick, S.E. and R.J. Hijmans (2017) WorldClim 2: new 1km spatial resolution climate surfaces for global land areas. *Int J Climatol* 37 (12): 4302-4315.

Supplementary Fig. 8 | Similarity between presence and absence points. Bars show correlation between climate (mean annual total precipitation, mean annual temperature) of presence and absence points across all combinations of forest biome and sub-continental window (n=70). Climate data is from WorldClim v2.1 (1970-2000) at 10 minute (~18 km) spatial resolution (Fick & Hijmans 2017).

R1.6

While I am glad to see the authors used more than 1 GCM, I don't consider 3 GCMs to be a particularly robust result. While there is a statement about "skill selected global climate models", I don't think the reference looked at VPD. Why not use more here? Typically for climate change assessment you want to use at least 10 GCM ensembles.

Response

Revised. L471-481, Supplementary Fig. 7. Thanks for raising this point and highlighting the need to more clearly present our rationale. We share Reviewer 1's concerns for maximising the robustness of the GCM data but we disagree that simply adding more GCMs will achieve this. We briefly discuss our approach in the context of climate model ensemble design to explain why.

Using a full ensemble of all available models is the gold standard for comprehensive projections (particularly where the quantity of interest is directly simulated by the model) and model evaluations, as found in IPCC reports, but is exceedingly rare in wildfire impact assessments. As an example there are 36 CMIP5 models which include historical, RCP4.5 and RCP8.5 data for air temperature, precipitation and specific humidity (<https://esgf-node.llnl.gov/projects/cmip5/>). An informal survey of around 20 wildfire projection studies yielded a maximum ensemble size of 23 (Hanan et al. 2021), with about half having 11-20 models and the other half having 1-10 models.

In the absence of a full ensemble, the most defensible approach in our view is using an objectively designed ensemble. This involves explicit selection of a subset of models from the full ensemble, following objective and transparent criteria that allow users to understand the implications of what has and has not been included. Objectively designed ensembles are commonly selected on the basis of model skill, model independence, or the ability of models to span the range of projected future change in climate – ideally all three (Evans et al. 2014; McSweeney et al. 2015). There is greater confidence in projections based on such ensembles due to the omission of poorly performing models, the lack of duplication of models with similar biases and the focus on the outer bounds of the distribution of likely changes in future climate, which are of great interest to decision makers including fire managers.

A third, less robust approach to climate projections is to use 'ensembles of opportunity'. Such ensembles vary widely in size and can be quite large (18, 19 and 23 member ensembles featured in three wildfire studies published in 2021), but generally lack a rationale for why some models are included and others excluded. Nevertheless, confidence in ensembles of opportunity can be improved by providing information on things like the skill, independence and fractional range of coverage of future climate space for those models included with respect to the full ensemble (McSweeney et al. 2015; Frieler et al. 2017). The number of GCMs used in a study is thus not by itself a reliable indicator of the robustness of results.

Regardless of which approach is taken, a necessary final step in any model selection process is restricting the ensemble to those models for which the required output data are available i.e. particular CMIP experiments, climate variables, time resolution, etc..

For our study we developed an objectively designed ensemble following the comprehensive global evaluation of CMIP5 models by McSweeney et al. 2015. We began with six models with a performance ranked as 'satisfactory' and output ranked as 'outliers' i.e. spanning the range of future changes in climate. We cross-referenced this with 12 GCMs for which 3-hourly data was available for historical, RCM4.5 and RCM8.5 experiments and for the specific humidity, air temperature and surface pressure variables required to calculate VPD. This resulted in a two member ensemble (ACCESS1-0 and CNRM-CM5). To increase the ensemble size we then expanded the search to a second set of six models from McSweeney et al. (2015) whose performance was ranked 'satisfactory' and whose output was not considered an outlier. Required output data were available for three of these models, however they were all variants of the GFDL model and were thus not independent. We selected the newest of the three models (GFDL-CM3), which has an improved representation of a number of atmospheric and land surface processes compared to the other two GFDL models.

To improve the robustness of our findings we see three main possibilities. One, relax the skill requirement and include all models for which 3-hourly data is available. This would generate a broader range of output, albeit a range featuring multiple models that have been flagged by McSweeney et al. (2015) as sufficiently implausible as to undermine confidence in their projections. Two, relax the 3-hourly data requirement and use coarser resolution data. As noted by Reviewer 1, the use of daily data is a key strength of this work (we have revised the manuscript to emphasise this) and using the highest possible time-resolution data (hourly data for reanalysis and 3-hourly data for GCMs) provides the most confidence in the robustness of our estimates of maximum daily VPD. Three, provide further information to help assess the robustness of the selected models in the context of the full set of CMIP5 models. This information shows that in most cases, the models we have included span most or all of the range of projected changes in climate. There is therefore reasonable confidence in the robustness of our projections. Based on this systematic review of model ensemble characteristics we concluded that the third option is preferable to the first two and refer to Supplementary Fig. 7 below, reproduced here. We have added selected items from this response to the Methods section to clarify model selection and provide further information about its robustness as follows:

“For climate change analyses we selected three global climate models from the CMIP5 dataset on the basis of skill, independence and the ability to span the range of future changes in climate: ACCESS1.0, CNRM-CM5 and GFDL-CM3 (Supplementary Table 5). These models were among the best performing compared to other CMIP5 models in a comprehensive evaluation for the purposes of downscaling over multiple regions, which included annual cycles of rainfall and temperature, general circulation patterns, teleconnections and the south east Asian monsoon. Of the highly performing models evaluated, these three models generally spanned all or most of the range of projected future seasonal and regional changes in climate (Supplementary Fig. 7). We avoided models from the same model family to avoid duplication of models with similar biases.”

References

Evans, J. P., Ji, F., Lee, C., Smith, P., Argueso, D., Fita, L. (2014) Design of a regional climate modeling projection ensemble experiment— NARCLiM. *Geoscientific Model Development*, 7, 621–629. <https://doi.org/10.5194/gmd-7-621-2014>.

Frieler, K., Lange, S., Piontek, F., Reyer, C. P. O., Schewe, J., Warszawski, L. et al. (2017). Assessing the impacts of 1.5 °C global warming: simulation protocol of the Inter-Sectoral Impact Model Intercomparison Project (ISIMIP2b). *Geoscientific Model Development*, 10(12), 4321-4345. <https://doi.org/10.5194/gmd-10-4321-2017>.

Hanan, E. J. et al. How climate change and fire exclusion drive wildfire regimes at actionable scales. (2021) *Environ. Res. Lett.* 16, 024051

McSweeney, C. F., Jones, R. G., Lee, R. W., and Rowell, D. P. (2015) Selecting CMIP5 GCMs for downscaling over multiple regions, *Climate Dynamics*, 44, 3237–3260. <https://doi.org/10.1007/s00382-014-2418-8>.

Supplementary Fig. 7 | Range of future climate change in selected climate models. Markers show seasonal mean temperature and precipitation in Europe, South East Asia (SEA) and Africa from the full CMIP5 model ensemble (adapted from Fig 11 in McSweeney et al. 2015). We selected three models based on skill, independence and the ability to span the range of future changes in climate. We have added coloured circles to the plot show the three models used in this study (red) and four additional models that also met selection criteria but for which model output was not available (blue). Considering actual (red) and potential (blue) models, the actual models span the full range of precipitation changes in 5 out of 12 cases, the full range of temperature changes in 7 out of 12 cases, at least 75% of the range of precipitation changes in 9 out of 12 cases and at least 75% of the range of temperature changes in 11 out of 12 cases.

R1.7

Each GCM will have biases though, so you'd ideally want to perform bias correction to get comparable VPD. The delta bias-correction approach for treating GCM data would be fine if the daily distributions of VPD from GCMs credibly represented those from ERA-5. It is unlikely that they are though. While delta bias-correction is OK for many climate change assessments, I am concerned here given that use of thresholds from ERA-5 data were used. One would ideally want to use a more sophisticated BC approach here to account for potential large differences in the distribution.

Response

Revised. L105-109, L110-113, L485-487, Fig. 3-5, Supplementary Fig. 4-6. We have bias corrected all global climate model data using a quantile mapping approach (Cannon et al. 2015) commonly used in wildfire research (Abatzoglou et al. 2019; Kirschmeier-Young et al. 2019; Ruffault et al. 2020). After bias correction the spatial pattern of results is broadly similar, the magnitude of increases is generally greater (particularly for the ACCESS1-0 model) but the overall conclusions are unchanged. The key text changes are as follows:

L105-109: “Under a high emissions scenario (RCP8.5), by 2026-2045 all models projected at least 30 45 additional days per year above the VPD threshold in parts of tropical South America, with two out of three models also projecting increases of this magnitude in North America, east Africa and large parts of Europe (Supplementary Fig. 4).”

L110-113: “Then VPD thresholds will be exceeded by at least 45 additional days per year in forest biomes on every continent, including increases of at least 150 days per year in tropical South America, regardless of model.”

All figures have been updated along with the methods as follows:

L485-487: “Daily ERA5 data (1981-2000) was used to bias correct GCM VPD following a quantile mapping approach.”

References

Abatzoglou, J. T., Williams, A. P., & Barbero, R. (2019). Global emergence of anthropogenic climate change in fire weather indices. *Geophysical Research Letters*, 46, 326–336.

<https://doi.org/10.1029/2018GL080959>

Cannon, A. J., Sobie, S. R., Murdock, T. Q. (2015). Bias correction of simulated precipitation by quantile mapping: How well do methods preserve relative changes in quantiles and extremes? *Journal of Climate*, 28:6938-6959. doi:10.1175/JCLI-D-1400754.1

Kirschmeier-Young, M. C., Gillett, N. P., Zwiers, F. W., Cannon, A. J., & Anslow, F. S. (2019). Attribution of the influence of human-induced climate change on an extreme fire season. *Earth's Future*, 7, 2–10.

<https://doi.org/10.1029/2018EF001050>

Ruffault, J., Curt, T., Moron, V., Trigo, R. M., Mouillot, F., Koutsias, N., Pimont, F., Martin StPaul, N., Barbero, R., Dupuy, J-L., Russo, A., Belhadj-Khedher, C. (2020) Increased likelihood of heat-induced large wildfires in the Mediterranean Basin. *Scientific Reports* 10, 13790. doi:10.1038/S41598-020-70069-Z

R1.8

Smoke impacts can spread well downwind of fires. In the absence of using a smoke dispersion model, it would be good to have strong justification for a distance from potential fires. There is reference to "GCM allowing smoke transport for tens of kilometers", which doesn't seem very logical as GCM resolutions are often 100-200km. Again, it might be useful to point to literature on wildfire smoke impacts here to justify choices for population exposure.

Response

Revised. L524-526. Thanks for raising this point. Population exposure to wildfire smoke is a complex function of many factors including fire intensity, weather conditions and plume injection height (Williamson et al. 2016). There are documented examples of smoke transport beyond the grid cell size of GCMs in Australia (Price et al. 2012; Di Virgilio et al. 2021), the U.S. (Kalashnikov et al. 2022) and southeast Asia (Koplitz et al. 2016), while a recent highly cited review noted air pollution effects have been found as far as 1000 km away (Xu et al. 2021).

Two of our three original references (Shaposhnikov et al. 2014; Johnston et al. 2021) did not explicitly refer to the long range aspect of smoke transport and we have replaced them with the aforementioned review (Xu et al. 2021) and revised the text as follows:

L524-526: “Because results are reported at the coarser resolution of GCM grid cells, they allow for long range smoke transport, which has been observed around the world at a scale of hundreds of kilometres or more.”

References

Di Virgilio, G., Hart, M.A., Maharaj, A.M., Jiang, N. (2021) Air quality impacts of the 2019–2020 Black Summer wildfires on Australian schools. *Atmospheric Environment*, 261, 118450, <https://doi.org/10.1016/j.atmosenv.2021.118450>.

Kalashnikov DA, Schnell JL, Abatzoglou JT, Swain DL, Singh D. (2022) Increasing co-occurrence of fine particulate matter and ground-level ozone extremes in the western United States. *Sci Adv*, 8(1), eabi9386. doi:10.1126/sciadv.abi9386.

Koplitz, S. N., Mickley, L. J., Marlier, M. E., Buonocore, J. J., Kim, P. S., Liu, T., Sulprizio, M. P., DeFries, R. S., Jacob, D. J., Schwartz, J., Pongsiri, M., and Myers, S. S. (2016) Public health impacts of the severe haze in Equatorial Asia in September–October 2015: demonstration of a new framework for informing fire management strategies to reduce downwind smoke exposure, *Environmental Research Letters*, 11, 094023. <https://doi.org/10.1088/1748-9326/11/9/094023>.

Price, O.F., Williamson, G.J., Henderson, S.B., Johnston, F. and Bowman, D.J.M.S. (2012) The relationship between particulate pollution levels in Australian cities, meteorological variables, and landscape fire activity detected from MODIS hotspots. *PLoS ONE* 7, e47327.

Williamson, G. J. et al. (2016) A transdisciplinary approach to understanding the health effects of wildfire and prescribed fire smoke regimes. *Environ. Res. Lett.* 11, 125009.

Xu R, Yu P, Abramson MJ, et al. Wildfires, global climate change, and human health. *N Engl J Med* 2020; 383: 2173–81.

R1.9

Did you use projected population density data or leave things at the 1990-2000 levels?

Response

Revised. Fig. 5, L32-34, L124-129, L205-207, L526-528. Thanks for raising this. We originally used current population levels but have updated our plots using “middle of the road” projections for 2090 drawn from the same dataset (Jones et al. 2016). The use of these projections led to some changes in the pattern of future population exposure to forest fire smoke as follows:

L32-34: “Escalating forest fire risk threatens catastrophic carbon losses in the Amazon and major population health impacts from wildfire smoke in south Asia and east Africa.”

L124-129: “Substantial increases in the number of days over the VPD threshold – and hence days of elevated probability of fire and smoke emissions – are projected to occur by 2081-2100 near major

population centres in south Asia and east Africa by all three models (Fig. 5). Two of three models also suggest considerable population exposure to smoke from increased forest fire activity in parts of central America, west Africa and east Asia.”

L205-207: “We also show that increases in forest fire activity are projected to occur near major population centres in east Africa and south Asia, and possibly central America, east Asia and west Africa.”

We have updated Fig. 5 and the methods text as follows:

L526-528: “Population projections for 2090 were based on a “middle of the road” scenario in terms of expected population growth, urbanization, and spatial patterns of development.”

References

Jones, B., & O’Neill, B. C. (2016). Spatially explicit global population scenarios consistent with the Shared Socioeconomic Pathways. *Environmental Research Letters*, 11(8), 084003.

R1.10

Fig 1: Perhaps include an inset map in each plot to show the biome of interest

Response

Revised. Fig. 1. Inset added.

R1.11

Fig 3-5: The colormap here is not ideal as light blue covers increases of up to 30-days. Consider using a colormap with white/grey centered around a change of 0.

Response

Revised. Fig. 3-5, Supplementary Fig. 4-6. White added to zero region of colormap.

Reviewer 2 Comment 1 (R2.1)

This is an interesting paper, but I do have concerns about whether or not the number of days exceeding the VPD threshold is actually related to area burned among the 70 regions analyzed. This is not presented, and when I qualitatively look at the results and compare them to my mental image of fire prone areas, I am not convinced there is a relationship.

A little less qualitative: in looking at the animation here (https://earthobservatory.nasa.gov/global-maps/MOD14A1_M_FIRE), Japan and the Korean peninsula have very little fire, yet according to figure 3a, it has by far the highest number of days exceeding the VPD threshold. The same can be said for parts of Europe such as Scandinavia.

It is not clear how to interpret the results pertaining to the number of days exceeding the VPD threshold. On the surface, I’d think that areas of the globe with high values (in fig. 3a) would be exceptionally fire prone and exhibit high amounts of area burned. However, I don’t consider Japan and the Korean peninsula particularly fire prone. Same goes for northern Europe. In the southeastern USA, the frequency of VPD exceedance is high, but most fires are prescribed fires.

Related and very important: if the frequency of days exceeding the VP threshold and area burned are not at least moderately correlated, and projections of effects to carbon or people under a future climate are potentially suspect. I guess I’d like to see some sort of analyses that relates the frequency of VPD exceedance to area burned (by sub-continental window, for example). If this relationship is moderately strong, then there is reason to make the projections under climate change.

Response

Revised. L182-185, L515-517, L517-519. We share Reviewer 2's concern that our results are robust and clearly communicated, so we have pooled several related comments about the relationship between the number of days that empirically-derived VPD thresholds are exceeded, and fire activity.

The heart of our paper is logistic regression modelling of the daily maximum VPD value (plus confidence intervals) at which the probability of fire – the probability of observing a burned area detectable on MODIS imagery – is greater than 0.5.

We show there is a strong relationship between VPD and fire activity in a wide range of fire-prone forest biomes around the world (Supplementary Fig. 1, Supplementary Table 1). Although there are some regions where this relationship is weak, the worst performing models represent about one seventh of the mean annual area burnt of the best performing models (Supplementary Note 1).

This relationship is derived on an individual forest biome / subcontinental window basis. There is considerable variation in the value of the VPD threshold within and between biome groups (subtropical to tropical, temperate and boreal, and mediterranean), as well as in the number of days over this threshold.

As suggested by the reviewer, we calculated the correlation between days over VPD threshold and total burned area for each combination of forest biome and sub-continental window, based on monthly area averages rather than individual daily pixels (Supplementary Fig. 9, reproduced below). These results are consistent with our main analysis i.e. the correlation is generally strong (r averaged across biomes is 0.39), but varies within biomes.

Importantly, it does not follow from any of our analyses that area burnt scales with the total number of days over VPD threshold. The strength of the relationship is not necessarily correlated with the magnitude of area burnt or the number of days over threshold for any given forest biome and sub-continental window.

On a related note, our analysis is guided by the four switch model (Archibald et al. 2009; Bradstock 2010), which argues that the fundamental biophysical preconditions for landscape fire are 1) sufficient fuel, 2) sufficiently low fuel moisture, 3) sufficient fire weather conditions and 4) an ignition source. A corollary of this model is that fire regimes vary in the identity of the switch that acts to limit overall fire incidence overall. As noted in the introduction, there is good evidence that fuel moisture plays an important role in limiting fire behaviour in forest biomes globally, given their abundance of fuel. It is likely, however, that in some circumstances ignition will limit forest fire activity even in the presence of sufficiently dry fuel. High population density, high detection rates and high suppression capacity are all known to lower the effective ignition rate (Collins et al. 2018; Clarke et al. 2019).

In summary, we have modelled fire activity as a function of VPD, but 1) there is not necessarily a relationship between the absolute number of days over VPD thresholds and the absolute amount of fire activity, 2) there are areas where the model does not perform well, and 3) there are areas where fuel moisture may not be the limiting factor on overall fire activity. We have made the following additions to the manuscript:

L182-185: “High population density, high fire detection rates and high suppression capacity are all known to lower the effective ignition rate and could weaken the link between VPD and fire activity in some regions.”

L515-517: “Note that the strength of the relationship between VPD and fire activity in any given region does not imply a particular magnitude of burnt area for a given number of exceedances of daily VPD threshold values.”

L517-519: “A supplemental analysis examined the relationship between area averaged monthly days over $VPDP=50$ and burnt area, with broadly similar findings to the main analysis (Supplementary Fig. 9).”

References

Clarke, H., Gibson, R., Cirulis, B., Bradstock, R. A., and Penman, T. D. (2019). Developing and testing models of the drivers of anthropogenic and lightning-caused wildfire ignitions in south-eastern Australia. *J. Environ. Manage* 235, 34–41. doi: 10.1016/j.jenvman.2019.01.055

Collins, K.M., Price, O.F., Penman, T.D. (2018) Suppression resource decisions are the dominant influence on containment of Australian forest and grass fires. *J Environ Manag*, 228, 373–82. <https://doi.org/10.1016/j.jenvman.2018.09.031>.

Supplementary Fig. 9 | The correlation between monthly area burnt and monthly frequency of VPD threshold exceedances (days). Each dot represents a combination of forest biome and sub-continental window, boxplots represent forest biomes and are shown in descending order by median days over VPD threshold (center line, median; box limits, upper and lower quartiles; whiskers, 1.5x interquartile range; points, outliers). Forest biomes are further classified into three biome groups. Biome legend is in Supplementary Table 3.

R2.2

I'm guessing that a lot of the fire seen in some parts of the planet are cultural, agricultural, or prescribed fires. If this is the case, can projections like this even be made? Related, for those fires that are cultural/agricultural/prescribed, they probably serve to stabilize carbon, meaning these fires are generally intended to preserve large trees and not kill them. So many fires in these areas are not necessarily a threat to carbon, now and into the future.

Response

Revised. L462-465. Thanks for raising this point. We do not consider agricultural burns, cultural burns or prescribed burns to have accounted for a significant proportion of our fire activity data. Prescribed burns are generally of far lower size and intensity than wildfires and are frequently undetected by MODIS (Chuvieco et al. 2020). Cultural burns are of even smaller magnitude than prescribed burns.

Given our focus on forests, we have used a forest mask (Schepaschenko et al. 2015; see Methods section L450-451) and consequently it is unlikely that agricultural fires represent a strong contribution. While cultural and prescribed burning may serve to stabilise carbon in some cases, the burns associated with logging in the Amazon and other tropical rainforest regions are less likely to do so. We have added the following text:

“Although they are of great interest, prescribed and cultural burns are not likely to have accounted for a significant proportion of the fire activity data as they are generally of far lower size and intensity than wildfires and are frequently undetected by MODIS⁶¹.”

References

Chuvieco, E., Aguado, I., Salas, J., García, M., Yebra, M., Oliva, P. (2020) Satellite remote sensing contributions to wildland fire science and management. *Curr. For. Rep.* 6, 81–96.

Schepaschenko, D., See, L., Lesiv, M., McCallum, I., Fritz, S., Salk, C., Moltchanova, E., Perger, C., Shchepashchenko, M., Shvidenko, A., Kovalevskiy, S., Gilitukha, D., Albrecht, F., Kraxner, F., Bun, A., Maksyutov, S., Sokolov, A., Durauer, M., Obersteiner, M., Karminov, V., & Ontikov, P. (2015). Development of a global hybrid forest mask through the synergy of remote sensing, crowdsourcing and FAO statistics. *Remote Sensing of Environment*, 162, 208-220. doi:10.1016/j.rse.2015.02.011

R2.3

Scientists are increasingly being criticized for exaggerating the effects of climate change by, for example, using the most extreme climate change scenarios in their analyses. It is my understanding that RCP 8.5 is unlikely, so it is perhaps more appropriate to use a more relevant emissions scenario for the main findings. I know it is less splashy, but I think it is important to not overexaggerate climate change effects in the abstract and the main figures in the paper. Sure, keep RCP 8.5, but that might be better in the supplemental as opposed to the main findings.

Response

Not revised. We agree that care needs to be taken in presenting results from realistic emissions scenario. We acknowledged in the original manuscript the increasing plausibility of RCP4.5 (L115-117: “Under a lower and increasingly more plausible emissions scenario (RCP4.5) the magnitude of change is smaller but still features widespread increases in the annual frequency of days of elevated probability of fire (Supplementary Fig. 5 and 6)”).

However, global CO₂ emissions rebounded by nearly 5% in 2021 (IEA, 2022) suggesting that a trajectory closer to RCP4.5 than RCP8.5 is not a fait accompli. In addition, it has recently been argued that worst case scenarios are underexplored (Kemp et al. 2022). On balance we argue that understanding the potential effects of RCP8.5 on forest fire risk is important enough to place this scenario in the main text, with further results in the Supplementary Information.

References

IEA (2020) Global Energy Review 2021, <https://www.iea.org/reports/global-energy-review-2021/co2-emissions>

Kemp, L., Xu, C., Depledge, J., Evi, K.L., Gibbins, G., Kohler, T.A., Rockstrom, J., Scheffer, M., Schellnhuber, H.J., Steffen, W., Lenton, T.M. (2022) Climate Endgame: Exploring catastrophic climate change scenarios. PNAS, 119(34), e2108146119.

R2.4

The absolute change in the number of days exceeding the VPD threshold under climate change is interesting, but I am wondering about a relative metric of projected change. For example, I can envision some parts of the boreal forest increasing from 10 to 30 days, which is 20 day increase but a three-fold increase. Will the implications of this differ when compared to a projected increase from 100 to 120 days in other parts of the globe?

Response

Revised. L519-521, Supplementary Figs. 10-13. Our original analysis included figures showing relative change but we omitted them as they did not substantially modify the overall conclusions. However, we agree that this analysis may be of interest to some readers and Supplementary Figs. 10-13 (reproduced below) show the relative change in days above VPD threshold by 2026-245 and 2081-2100 under RCP8.5 and RCP4.5 for each GCM. Text has been added to the methods as follows:

“A supplemental analysis examined the relative, rather than absolute, change in the number of days over VPD=50, with broadly similar findings (Supplementary Figs. 10-13).”

Supplementary Fig. 10 | Projected relative change in the mean annual frequency of days exceeding the VPD thresholds by 2026-2045 under RCP8.5 for the GFDL-CM3 (a), CNRM-CM5 (b) and ACCESS1.0 (c) models. To resolve the majority of the distribution of relative change values, outliers are minimised by capping the scale at 800% (the highest value is 4600%) and omitting pixels with fewer than 5 days per year currently exceeding VPD thresholds.

Supplementary Fig. 11 | Projected relative change in the mean annual frequency of days exceeding the VPD thresholds by 2081-2100 under RCP8.5 for the GFDL-CM3 (a), CNRM-CM5 (b) and ACCESS1.0 (c) models. To resolve the majority of the distribution of relative change values, outliers are minimised by capping the scale at 800% (the highest value is 4600%) and omitting pixels with fewer than 5 days per year currently exceeding VPD thresholds.

Supplementary Fig. 12 | Projected relative change in the mean annual frequency of days exceeding the VPD thresholds by 2026-2045 under RCP4.5 for the GFDL-CM3 (a), CNRM-CM5 (b) and ACCESS1.0 (c) models. To resolve the majority of the distribution of relative change values, outliers are minimised by capping the scale at 800% (the highest value is 4600%) and omitting pixels with fewer than 5 days per year currently exceeding VPD thresholds.

Supplementary Fig. 13 | Projected relative change in the mean annual frequency of days exceeding the VPD thresholds by 2081-2100 under RCP4.5 for the GFDL-CM3 (a), CNRM-CM5 (b) and ACCESS1.0 (c) models. To resolve the majority of the distribution of relative change values, outliers are minimised by capping the scale at 800% (the highest value is 4600%) and omitting pixels with fewer than 5 days per year currently exceeding VPD thresholds.

REVIEWERS' COMMENTS

Reviewer #1 (Remarks to the Author):

The authors did a nice job addressing my initial round of comments on the paper. I am adding a few additional notes that may warrant modifications:

- 1) Abstract, consider providing context to the term unprecedented as some of these regions had greater burned area prior to colonization.
- 2) Also potentially relevant to highlight Balch et al. 2022 who identifies VPD thresholds for flammability across biomes and continents in a similar fashion to what is done here.
- 3) Fig 3 suggestions: (a) mask out non forested land in a grey shade otherwise it looks white as in no change; (b) the legend should include the source of the data, specifically that ERA5 is used in panel (a), but GCMs in the other three panels
- 4) Fig 5; this is a lot of empty real estate. This figure could be taken to show disingenuously that there is "no" change in risk to population in many regions. Can you better show impact here by changing the contours? Also, the units are odd, person days per year per 1000 km²? As stated the units are lacking. Finally, the colourbar used in these figures is no appropriate per my original comment. The light blue indicates increased risk separated by white = no change and dark blue = decrease.
- 5) Since the authors insist on using the three GCMs which they articulated a nice argument for in the response, I think it is still necessary to provide some context for the three individual GCMs when showing results (e.g., Fig 3b-d). For example are the differences across models highlighting something about different climate sensitivities or regional changes in precipitation? Alternatively, if you do not wish to discuss model differences, consider showing results for individual models in the SI and show the 3-model mean change in the main paper.

References

Balch, J.K., et al. Warming weakens the night-time barrier to global fire. *Nature* 602, 442–448 (2022). <https://doi.org/10.1038/s41586-021-04325-1>

Response to reviewer comments – Forest fire threatens global carbon sinks and population centres under rising atmospheric water demand

Note that line numbers quoted in the response refer to the manuscript with tracked changes showing all markup.

Reviewer 1 Comment 1 (R1.1)

Abstract, consider providing context to the term unprecedented as some of these regions had greater burned area prior to colonization.

Response

Revised. L23-25. The opening sentence now reads: “Levels of fire activity and severity that are unprecedented in the instrumental record have recently been observed in the western US, south-eastern Australia, Siberia and other forested regions around the world.”

R1.2

Also potentially relevant to highlight Balch et al. 2022 who identifies VPD thresholds for flammability across biomes and continents in a similar fashion to what is done here.

Response

Revised. L173-175. We have added the following sentence: “A recent study identified VPD thresholds associated with fire activity in North and South America between 2017-2020 using hourly data, with similar findings³⁷.”

37. Balch, J.K., Abatzoglou, J.T., Joseph, M.B. et al. Warming weakens the night-time barrier to global fire. *Nature* 602, 442–448 (2022). <https://doi.org/10.1038/s41586-021-04325-1>

R1.3

Fig 3 suggestions: (a) mask out non forested land in a grey shade otherwise it looks white as in no change; (b) the legend should include the source of the data, specifically that ERA5 is used in panel (a), but GCMs in the other three panels

Response

Revised. Fig 3. Thanks for these suggestions. Following Balch et al. 2022 we have changed the colour corresponding to values close to zero from white to grey. We have done the same for Figures 2, 4 and 5 and related Supplementary Figures. We have updated all legends with: “The white areas indicate non-forest land.” We have added a reference to ERA5 in the Figure 3 legend as follows:

“Fig. 3 | The mean annual frequency of daily VPD threshold exceedances (days) for global forest biomes. Current frequency based on ERA5 data (2003-2020) (a) and the projected change in the number of days over VPD threshold by 2081-2100 under RCP8.5 for the GFDL-CM3 (b), CNRM-CM5 (c) and ACCESS1.0 (d) models.”

R1.4

Fig 5; this is a lot of empty real estate. This figure could be taken to show disingenuously that there is "no" change in risk to population in many regions. Can you better show impact here by changing the contours? Also, the units are odd, person days per year per 1000 km²? As stated the units are lacking. Finally, the

colourbar used in these figures is no appropriate per my original comment. The light blue indicates increased risk separated by white = no change and dark blue = decrease.

Response

Revised. Fig 5. Thanks for these suggestions. We appreciate your concern about implying that there is no change in risk for population in many regions. The distribution of changes is skewed and it would take a log scale or similar to bring out the lower positive changes, which would then greatly understate the magnitude of the large changes. On balance we believe it is more important to highlight the latter.

We agree that the units are odd, but this is because they represent an unfamiliar quantity. They are accurate and in the absence of an alternative suggestion we believe they are the simplest and best option for conveying the concept of joint risk of large populations exposed to increased forest flammability.

We have changed the light blue to yellow to make it easier to tell substantial positive from negative change.

R1.5

Since the authors insist on using the three GCMs which they articulated a nice argument for in the response, I think it is still necessary to provide some context for the three individual GCMs when showing results (e.g., Fig 3b-d). For example are the differences across models highlighting something about different climate sensitivities or regional changes in precipitation? Alternatively, if you do not wish to discuss model differences, consider showing results for individual models in the SI and show the 3-model mean change in the main paper.

Response

Revised. L122-126. We have added context for differences between the individual GCMs in the discussion of results as follows:

“The increases are greatest and most widespread in ACCESS1-0 and GFDL-CM3 and generally more moderate in CNRM-CM5. The latter tends to project the least warming of the three models, with GFDL-CM3 projecting the most²⁸. ACCESS1-0 is generally the driest of the three models, while both ACCESS1-0 and GFDL-CM3 have a higher climate sensitivity parameter than CNRM-CM5²⁹.”

28. McSweeney, C. F., Jones, R. G., Lee, R. W., & Rowell, D. P. (2015). Selecting CMIP5 GCMs for downscaling over multiple regions. *Climate Dynamics*, 44(11-12), 3237-3260. doi:10.1007/s00382-014-2418-8

29. Flato, G., J. Marotzke, B. Abiodun, P. Braconnot, S.C. Chou, W. Collins, P. Cox, F. Driouech, S. Emori, V. Eyring, C. Forest, P. Gleckler, E. Guilyardi, C. Jakob, V. Kattsov, C. Reason and M. Rummukainen, 2013: Evaluation of Climate Models. In: *Climate Change 2013: The Physical Science Basis. Contribution of Working Group I to the Fifth Assessment Report of the Intergovernmental Panel on Climate Change* [Stocker, T.F., D. Qin, G.-K. Plattner, M. Tignor, S.K. Allen, J. Boschung, A. Nauels, Y. Xia, V. Bex and P.M. Midgley (eds.)]. Cambridge University Press, Cambridge, United Kingdom and New York, NY, USA